# C2AL: Cohort-Contrastive Auxiliary Learning for Large-scale Recommendation Systems

## Abstract

Training large-scale recommendation models under a single global objective implicitly assumes homogeneity across user populations. However, real-world data are composites of heterogeneous cohorts with distinct conditional distributions. As models increase in scale and complexity and as more data is used for training, they become dominated by central distribution patterns, neglecting head and tail regions. This imbalance limits the model's learning ability and can result in inactive attention weights or dead neurons. In this paper, we reveal how the attention mechanism can play a key role in factorization machines for shared embedding selection, and propose to address this challenge by analyzing the substructures in the dataset and exposing those with strong distributional contrast through auxiliary learning. Unlike previous research, which heuristically applies weighted labels or multi-task heads to mitigate such biases, we leverage partially conflicting auxiliary labels to regularize the shared representation. This approach customizes the learning process of attention layers to preserve mutual information with minority cohorts while improving global performance. We evaluated C2AL on massive production datasets with billions of data points each for six SOTA models. Experiments show that the factorization machine is able to capture fine-grained user–ad interactions using the proposed method, achieving up to a 0.16% reduction in normalized entropy overall and delivering gains exceeding 0.30% on targeted minority cohorts.

## 1 Introduction

Deep neural networks (DNNs) have demonstrated substantial advancements across a variety of application domains, including computer vision (He et al., 2016), natural language processing (Vaswani et al., 2017), and classification (Krizhevsky et al., 2012) tasks. Recently, DNNs have served as the backbone of large-scale ads recommendation systems (Cheng et al., 2016), especially in computational advertising, where they are trained to predict user actions—such as clicks or conversions—by optimizing a global objective across massive datasets.

Although substantial research efforts (Caruana, 1997; Ma et al., 2018; Tang et al., 2020; Sener & Koltun, 2018; Liu et al., 2024; Yu et al., 2020; Jeong & Yoon, 2025) have focused on improving model prediction accuracy under the assumption of homogeneous training data, real-world production datasets are typically composed of latent sub-distributions, or cohorts, which complicates the learning process. This challenge becomes even more pronounced as model and dataset scales increase; optimization tends to favor the high-density regions of the data, thereby prioritizing majority cohorts (Crawshaw, 2020). As a result, models frequently underfit the distributional tails, failing to capture meaningful feature representations for minority cohorts. Particularly, in deep learning-based recommendation systems, when gradient updates are predominantly influenced by majority cohorts, the resulting attention mechanism (Xiao et al., 2017) tends to overemphasize interactions that are common within these groups, leaving many potential interaction pathways underutilized. This dynamic introduces representation bias—a systematic limitation in the model's ability to capture patterns unique to minority groups. Such bias results in poor generalization and miscalibration, ultimately reducing overall ad value and degrading the user experience.

To address this representation bias, we investigate auxiliary learning (Caruana, 1997), where secondary tasks are used during training to help improve the shared representation and are discarded at

inference. While empirically effective, the design of these tasks is often heuristic, and the mechanism by which they operate remains largely unanalyzed. We introduce Cohort-Contrastive Auxiliary Learning (C2AL)[1], a method for constructing auxiliary tasks that improve the training process via auxiliary losses, leading to active interaction between features in models and robust shared parameters.

Our analysis of the resulting learning dynamics reveals how C2AL systematically reshapes the model's factorization machine-based attention layer. Specifically, we demonstrate through gradient and parameter analysis that the C2AL objective compels the model's attention mechanism within modern architectures like the Deep and Hierarchical Ensemble Network (DHEN) (Zhang et al., 2022) to learn a statistically denser and less concentrated weight distribution, which was heavily influenced by the majority cohort to collapse toward modeling only high-density regions of the input space previously. This establishes an interpretable between the training objective, attention mechanism, and the learned representation; C2AL induces a more expressive representation by promoting a richer set of feature interactions, thereby strengthening its ability to generalize across heterogeneous cohorts in the data distribution.

Our contributions are twofold:

1. We introduce Cohort-Contrastive Auxiliary Learning (C2AL), a method for constructing targeted auxiliary tasks that mitigate representation bias in large-scale recommendation systems, leading to incremental overall performance improvement with no additional inference cost.

2. We carried out theoretical analysis and architecture research that explains the interpretability of C2AL, which reshapes the model's attention weights, resulting in denser and less concentrated feature interactions. By validating the interpretability on six production ads models with massive real-world data, C2AL demonstrating consistent pattern and significant improvements in predictive accuracy and normalized entropy.

## 2 RELATED WORKS

Our work uses the principles of Multi-Task Learning (MTL) as a framework for achieving mechanistic interpretability in large-scale recommendation models.

The central premise of MTL is to improve generalization by learning a shared representation across multiple tasks (Caruana, 1997). Our work focuses on a subset of MTL, auxiliary learning, where secondary training tasks serve as a regularization mechanism to shape a model's inductive bias for a *single* primary objective (Jaderberg et al., 2016). A key challenge in this paradigm is managing the interplay between task signals. This has been addressed through architectural solutions that learn task-specific parameter sharing (e.g., MMOE (Ma et al., 2018), PLE (Tang et al., 2020)) and by framing MTL as a multi-objective optimization problem (Sener & Koltun, 2018), which has led to algorithms that manage conflicting gradients and task priorities (e.g., PCGrad (Yu et al., 2020), CAGrad (Liu et al., 2024), and selective task updates (Jeong & Yoon, 2025)). These methods, however, are primarily designed to optimize joint task performance.

Much of the work in auxiliary learning has focused on developing effective, but often heuristic, methods to determine when a task is beneficial. For instance, the cosine similarity between task gradients can be used as a dynamic signal to gate the auxiliary loss and prevent negative transfer (Du et al., 2020). The prevailing intuition is that such methods constrain optimization to a more structured solution space, forcing the model to learn more generalizable representations (Nakhleh et al., 2024; Lippl & Lindsey, 2024). This is particularly relevant for avoiding spurious correlations that fail on minority cohorts with different underlying causal patterns (Hu et al., 2022; Li et al., 2023). However, these explanations often lack a concrete, mechanistic link to specific architectural components; they posit that representations become "better" but rarely demonstrate *how* or *where* this improvement is realized. Our work moves beyond these accounts by providing a precise, mechanistic explanation, linking a targeted auxiliary objective to a specific architectural change.

---

[1]The term "contrastive" refers to partially conflict traffic pattern *between cohorts* and is algorithmically unrelated to instance-discrimination objectives common in self-supervised learning (e.g., InfoNCE).

## 3 METHODOLOGY

### 3.1 PROBLEM SETUP

We consider a standard supervised learning setup for a primary prediction task, such as click pre-diction. Given an input feature vector $\mathbf{x} \in \mathcal{X}$, the goal is to predict a label $y \in \{0, 1\}$. Our model consists of two components: a shared representation encoder $f : \mathcal{X} \to \mathbb{R}^d$, parameterized by $\theta_S$, which maps the input to a $d$-dimensional embedding $\mathbf{h} = f(\mathbf{x}; \theta_S)$; and a primary prediction head $g_{\text{primary}} : \mathbb{R}^d \to [0, 1]$, parameterized by $\theta_H$, which produces the final prediction $\hat{y} = g_{\text{primary}}(\mathbf{h}; \theta_H)$.

The baseline single-task objective is to find the optimal parameters $\theta_S^*$ and $\theta_H^*$ that minimize the expected loss over the data distribution $\mathcal{D}$:

$$\{\theta_S^*, \theta_H^*\} = \arg \min_{\theta_S, \theta_H} \mathbb{E}_{(\mathbf{x},y)\sim\mathcal{D}} \Big[ \mathcal{L}(\hat{y}, y) \Big] \tag{1}$$

During back-propagation, both $\theta_S$ and $\theta_H$ will be updated according to the loss:

$$\theta^{(t+1)} = \theta^{(t)} - \alpha^{(t)} \nabla_\theta \mathcal{L}(\theta), \, \theta \in \{\theta_S, \theta_H\} \tag{2}$$

When optimized over heterogeneous data, this objective leads to the representation $\mathbf{h}$ becoming biased toward majority cohorts.

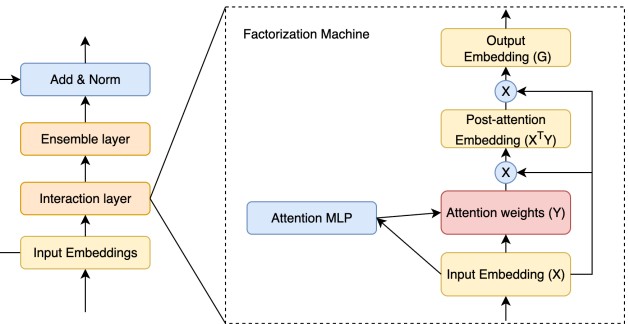

Figure 1: The Deep and Hierarchical Ensemble Network (DHEN) and its internal interaction layer.

The baseline models are built on the state-of-the-art architecture of computational ads recommen-dation systems: Deep and Hierarchical Ensemble Network (DHEN) (Zhang et al., 2022). A key component of this architecture is the interaction layer, which, in our case, is a factorization ma-chine (FM) based attention mechanism as shown in Fig. 1. The FM-based models compute pairwise feature interactions before projecting them into a compressed, computationally tractable space.

For a mini-batch, let $\mathbf{X} \in \mathbb{R}^{d \times m}$ be the matrix whose columns are the $m$ active $d$-dimensional sparse embeddings. The attention mechanism uses a weight matrix $\mathbf{Y} \in \mathbb{R}^{d \times k}$ to produce a compressed interaction embedding $\mathbf{G} \in \mathbb{R}^{d \times k}$ via the bi-linear form:

$$\mathbf{G} = \mathbf{X}\mathbf{X}^\top\mathbf{Y} \tag{3}$$

The term $\mathbf{X}\mathbf{X}^\top \in \mathbb{R}^{d \times d}$ represents the outer product of the batch's sparse features, and the learned attention matrix $\mathbf{Y}$ projects these interactions into a $k$-dimensional compressed space.

We visualized the weight distribution generated by the attention MLP. As shown in Figure 2, the attention weights of the baseline exhibit a light-tailed distribution: only a small subset of tokens receive meaningful values, while the majority of values are concentrated near zero. This observation indicates inefficient interactions within the input features/embedding, leading to representation bias which hurts model's performance, especially for under-represented segments.

### 3.2 THE C2AL FRAMEWORK

C2AL is a framework for mitigating this representation bias through two stages: (1) data-driven discovery of cohorts where a baseline model exhibits high predictive contrast, and (2) formulation of cohort-specific auxiliary tasks to regularize the shared parameters $\theta_S$.

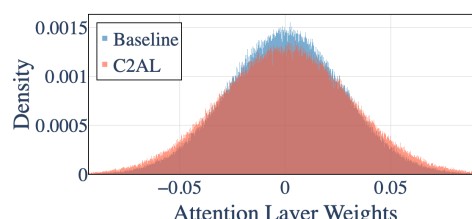 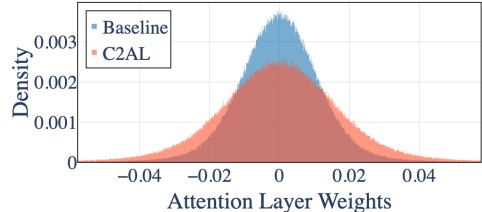

Figure 2: Comparison of attention weight sparsity between two representative production models. The baseline model demonstrates pronounced sparsity in attention weights, indicating that feature selection is primarily driven by majority cohorts. Consequently, features relevant to minority cohorts are frequently underrepresented or ignored.

### 3.2.1 CONTRASTIVE COHORT DISCOVERY

We segment the data along interpretable semantic axes (e.g., user value, age) into disjoint cohorts $\{\mathcal{C}_1, \ldots, \mathcal{C}_N\}$. For each axis, we use a baseline model's predictions to quantify pairwise divergence between cohort distributions, employing metrics such as KL divergence, Cosine Similarity, Jensen-Shannon distance, and Wasserstein distance.

We denote the pair of cohorts with maximal distributional disparity as $\mathcal{C}_{\text{head}}$ and $\mathcal{C}_{\text{tail}}$, and use them for auxiliary task construction. Figure 3 illustrates an example of the distributional divergence between these cohorts. Distributional divergence provides a principled, though not exclusive, criterion for cohort discovery; practical factors like cohort size and causal structure may also influence selection.

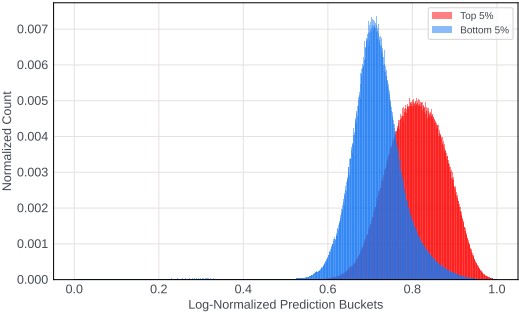

Figure 3: Probability density functions of the baseline Model A's (optimize for click objective on Instagram) predictions for the head and tail cohorts along a selected semantic axis, illustrating distributional divergence.

### 3.2.2 CONTRASTIVE AUXILIARY LEARNING

Given the identified cohorts, we construct two auxiliary binary classification tasks designed to inject targeted gradient signals into the shared encoder. We define two auxiliary labels, $y_{\text{head}}$ and $y_{\text{tail}}$, which are positive *only* for samples that are both positive for the primary task ($y = 1$) and belong to the corresponding cohort. Using the indicator function $\mathbb{I}(\cdot)$, this is expressed as:

$$y_{\text{head}} = y \cdot \mathbb{I}(\mathbf{x} \in \mathcal{C}_{\text{head}}) \qquad \text{and} \qquad y_{\text{tail}} = y \cdot \mathbb{I}(\mathbf{x} \in \mathcal{C}_{\text{tail}})$$

We augment the model with two auxiliary heads, $g_{\text{head}}$ and $g_{\text{tail}}$, parameterized by $\theta_{\text{head}}$ and $\theta_{\text{tail}}$, which take the shared representation $\mathbf{h}$ as input. The complete C2AL objective, optimized during training, is:

$$\mathcal{L}_{\text{C2AL}} = \underbrace{\mathcal{L}(g_{\text{primary}}(\theta_S; \theta_{\text{H}}), y)}_{\text{Primary Task Loss}} + \underbrace{\lambda_{\text{head}}\mathcal{L}\left(g_{\text{head}}(\theta_S; \theta_{\text{head}}), y_{\text{head}}\right) + \lambda_{\text{tail}}\mathcal{L}\left(g_{\text{tail}}(\theta_S; \theta_{\text{tail}}), y_{\text{tail}}\right)}_{\text{Cohort-Contrast Losses}} \quad (4)$$

Importantly, this objective is used only during training. At inference time, the auxiliary heads and their parameters $\{\theta_{\text{head}}, \theta_{\text{tail}}\}$ are discarded. The model reverts to the single-task architecture defined in Section 3.1 and is evaluated using Equation 1, incurring no additional computational cost or architectural changes at serving.

**Learning Dynamics.**    To analyze the learning dynamics, we simplify the C2AL training objective to a primary loss and a single weighted auxiliary loss, both functions of the embedding $\mathbf{G}$:

$$\mathcal{L}_{\text{C2AL}} = \mathcal{L}_{\text{primary}}(\mathbf{G}, y) + \lambda_{\text{aux}}\mathcal{L}_{\text{aux}}(\mathbf{G}, y_{\text{aux}}) \tag{5}$$

Since the FM-based attention mechanism plays a critical role in terms of embedding interaction and representation quality, we present how C2AL affects the attention module by taking the partial derivative with respect to the attention matrix $\mathbf{Y}$. Backpropagating to the attention matrix $\mathbf{Y}$, the gradient is given by:

$$\nabla_{\mathbf{Y}}\mathcal{L}_{\text{C2AL}} = (\mathbf{X}\mathbf{X}^{\top})(\nabla_{\mathbf{G}}\mathcal{L}_{\text{primary}} + \lambda_{\text{aux}}\nabla_{\mathbf{G}}\mathcal{L}_{\text{aux}}) \tag{6}$$

This equation provides the central mechanistic insight. When trained solely on the main objective (the baseline case, where $\lambda_{\text{aux}} = 0$), the gradient $\nabla_{\mathbf{Y}}\mathcal{L}_{\text{C2AL}}$ is dominated by majority cohorts, causing $\mathbf{Y}$ to converge to a sparse, concentrated state that captures only globally predictive feature interactions. The auxiliary term, however, injects cohort-specific gradient signals ($\nabla_{\mathbf{G}}\mathcal{L}_{\text{aux}}$) directly into the update for $\mathbf{Y}$. Since the compressed embedding $\mathbf{G}$ depends linearly on $\mathbf{Y}$, these targeted changes to the attention matrix map directly to changes in the representation:

$$\Delta\mathbf{G} = (\mathbf{X}\mathbf{X}^{\top})\Delta\mathbf{Y} \tag{7}$$

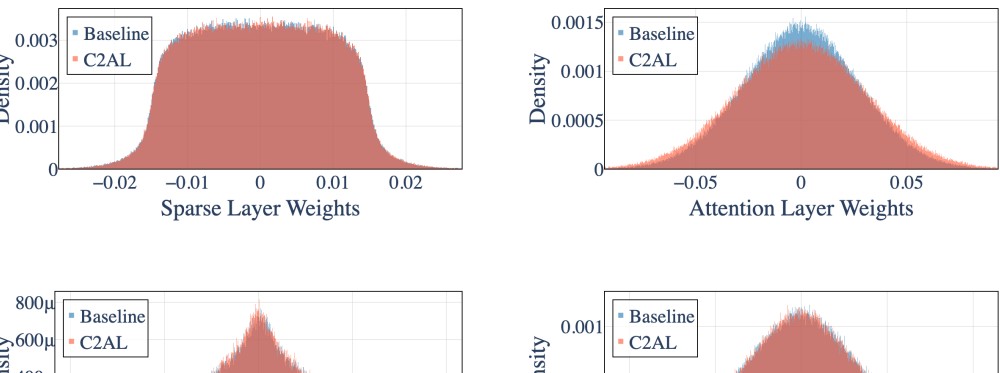

Figure 4: Weight distributions across various network layers. Left: two layers preceding the attention MLP. Right: attention weights (top) and post-attention weights (bottom).

To verify the impact of $\mathcal{L}_{\text{C2AL}}$ on attention weights $\mathbf{Y}$ and shared embedding $\mathbf{G}$, we first compare the parameter distribution differences between the baseline and our proposed method layer by layer. We found that the attention matrix weights changed significantly after applying C2AL, which aligns with our understanding that the FM plays an important role in shared embedding. Figure 4 demonstrates the weight distribution difference of layers before and after the attention MLP. From this visualization, we show that the shared attention weights had changed little in layers before the attention MLP, while changing greatly during the attention layer.

Training with C2AL produces an attention matrix with a visibly higher entry-wise diversity. This distribution difference is consistent across all the models evaluated in our experiments. Figure 2

shows two examples of distribution difference. This shift in the distribution of $\mathbf{Y}$ provides two benefits. First, a denser $\mathbf{Y}$ enables richer feature utilization, allowing a wider set of sparse embeddings to participate in meaningful second-order interactions. Second, a higher-diversity $\mathbf{Y}$ allows the model to more strongly weight the specific, and potentially rare, feature interactions that are critical for discriminating minority cohorts. These effects collectively yield a compressed representation $\mathbf{G}$ that is more expressive and cohort-aware, thereby improving the performance of the downstream primary task.

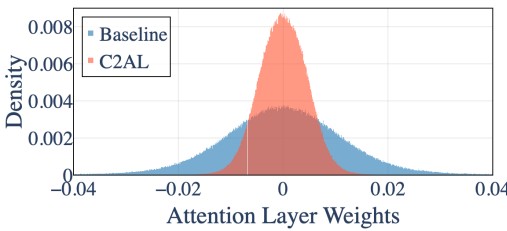

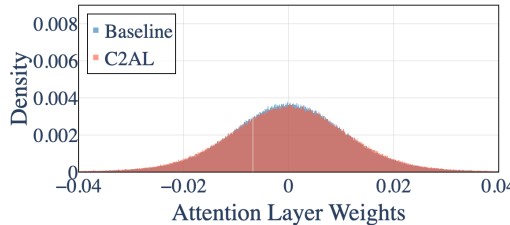

(a) At 0.4B training samples, the C2AL model is tightly concentrated, with most weights near zero.

(b) At 2.4B training examples, the C2AL model begins showing increased diversity and more non-zero weights.

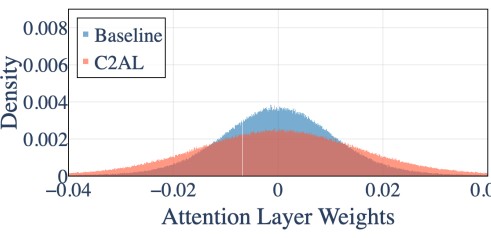

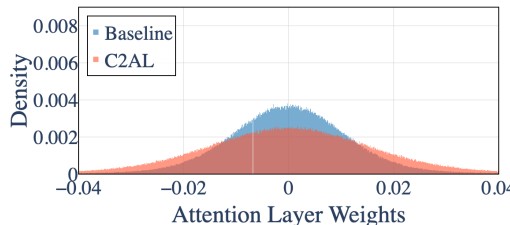

(c) At 7.2B training examples, the C2AL model displays higher diversity than the baseline.

(d) At 12B training examples, baseline distribution is unchanged and C2AL reaches max diversity.

Figure 5: Evolution of Attention Weights Throughout Training

**Gradient Dynamics.** The efficacy of C2AL arises from the partially conflicted task labels it introduced. Consider a positive sample ($y = 1$) from three data regions (we can similarly derive the C2AL labels for $y = 0$):

- If $\mathbf{x} \in \mathcal{C}_{\text{head}}$: The labels are $(y, y_{\text{head}}, y_{\text{tail}}) = (1, 1, 0)$. The gradients from $\mathcal{L}_{\text{primary}}$ and $\mathcal{L}_{\text{head}}$ align to update $\theta_S$, amplifying the learning signal for this minority cohort.

- If $\mathbf{x} \in \mathcal{C}_{\text{tail}}$: The labels are $(y, y_{\text{head}}, y_{\text{tail}}) = (1, 0, 1)$, creating a symmetric effect that amplifies the signal for the tail cohort.

- If $\mathbf{x} \notin \{\mathcal{C}_{\text{head}} \cup \mathcal{C}_{\text{tail}}\}$ (i.e., a majority sample): $(y, y_{\text{head}}, y_{\text{tail}}) = (1, 0, 0)$ The gradient from $\mathcal{L}_{\text{primary}}$ pushes $\mathbf{h}$ to be predictive of a positive outcome. Simultaneously, gradients from both auxiliary losses push $\mathbf{h}$ to be predictive of a *negative* outcome for their respective heads.

Given the gradient vector with respect to the shared parameters $G(\theta)$, $\theta_S$ is updated by:

$$\theta_S^{(t+1)} = \theta_S^{(t)} - \alpha^{(t)}\left(G_{\text{primary}}^{(t)} + G_{\text{aux}}^{(t)}\right) \tag{8}$$

where

$$G_{\text{primary}}(\theta_S) = \nabla_{\theta_S}\mathcal{L}(\theta_S, \theta_H, y) \tag{9}$$

$$G_{\text{aux}}(\theta_S) = \lambda_{\text{head}}\nabla_{\theta_S}\mathcal{L}(\theta_S, \theta_{\text{head}}, y_{\text{head}}) + \lambda_{\text{tail}}\nabla_{\theta_S}\mathcal{L}(\theta_S, \theta_{\text{tail}}, y_{\text{tail}}) \tag{10}$$

As described above, $y$, $y_{\text{head}}$, and $y_{\text{tail}}$ may be consistent for some of the samples. For example, $y_i = y_{\text{head},i}$ if $x_i \in \mathcal{C}_{\text{head}}$. In this case, the gradient vector fields $\nabla_{\theta_S} \mathcal{L}(\theta_S, \theta_H, y)$ and $\nabla_{\theta_S} \mathcal{L}(\theta_S, \theta_{\text{head}}, y_{\text{head}})$ satisfy

$$\langle \nabla_{\theta_S} \mathcal{L}(\theta_S, \theta_H, y), \nabla_{\theta_S} \mathcal{L}(\theta_S, \theta_{\text{head}}, y_{\text{head}}) \rangle = 1 \tag{11}$$

We can then define the projection and orthogonal components of $G_{\text{aux}}$ with respect to $G_{\text{primary}}$:

$$G_{\text{aux}}^{\parallel} := \frac{\langle G_{\text{aux}}, G_{\text{primary}} \rangle}{\|G_{\text{primary}}\|_2^2} \cdot G_{\text{primary}} \qquad \text{and} \qquad G_{\text{aux}}^{\perp} := G_{\text{aux}} - G_{\text{aux}}^{\parallel}$$

The learning rule in Equation 8 can thus be rewritten as:

$$\theta_S^{(t+1)} = \theta_S^{(t)} - \alpha^{(t)} \left( G_{\text{primary}}^{(t)} + G_{\text{aux}}^{\parallel(t)} + G_{\text{aux}}^{\perp(t)} \right) \tag{12}$$

Here, $G_{\text{primary}}^{(t)} + G_{\text{aux}}^{\parallel(t)}$ together drive convergence to a local minimum of the primary task loss (for sufficiently small $\alpha^{(t)}$), while $G_{\text{aux}}^{\perp(t)}$ acts as a regularization term, preventing the learning process from being trapped in local minima. Conventional regularization methods, such as $\ell_1$ and $\ell_2$ penalties, operate directly on a model's parameter space. C2AL, in contrast, imposes a *functional* regularization by constraining the model's predictive behavior on specific data cohorts. For a majority-cohort sample that is positive for the primary task, the auxiliary losses provide a counter-signal to the primary loss. To resolve this, the shared encoder must learn representations that are not merely predictive of the main label but are also sufficiently discriminative to distinguish majority samples from those belonging to the contrastive cohorts.

Figure 5 illustrates the evolution of attention weights throughout the training process of C2AL model versus its baseline. Although both models are initialized to a light-tail bell-shape distribution (C2AL model was initialized with even higher sparsity), as training progresses, our proposed method gradually learns more meaningful interactions between input embedding pairs from Equation 12. Therefore, the C2AL model converges to a distribution characterized by a higher concentration of non-zero attention weights, reflecting its improved ability to capture informative dependencies.

## 4 EXPERIMENTS AND RESULTS

We validate C2AL on six large-scale, production ads models. The scale and diversity of these systems provide a robust test for the effectiveness and generalizability of our method. In this section, we share our experimental results as well as a deep dive into the C2AL mechanism based on our observations.

### 4.1 DATASETS

Our models are trained on datasets in which each sample corresponds to a single ad impression. These models span several axes of variation common to industrial-scale ads systems:

- **Ranking Funnel Stage:** We evaluate models from both the computationally constrained early-stage and the high-fidelity final-stage of ranking cascade. Recommendation systems use a multi-stage cascade to balance predictive accuracy with computational latency, refining millions of candidate ads down to a few using models of increasing complexity.
- **Optimization Objective:** We evaluate C2AL on both models covering both click (CTR) and conversion (CVR) predictions.
- **Platform and Surface Type:** The models operate across Facebook (FB) and Instagram (IG), handling both onsite conversions, which occur within the platform's ecosystem, and offsite conversions on external advertiser domains, which present distinct data challenges.

A core challenge these global-scale models face is significant data heterogeneity. As shown in Table 1, the Positive Label Ratio (PLR) can vary by nearly five-fold between the head and tail cohorts of a single semantic axis. This pronounced behavioral divergence motivates the need for C2AL, which introduces a cohort-aware regularization.

Table 1: Examples of Positive Label Ratio and NE improvement across semantic axes.

| Semantic Axis | Head PLR | Tail PLR | Overall NE$_{\text{diff}}$ | Head NE$_{\text{diff}}$ | Tail NE$_{\text{diff}}$ |
|---|---|---|---|---|---|
| Revenue | 0.14% | 0.03% | -0.28% | -0.25% | -0.17% |
| Age | 0.05% | 0.04% | -0.14% | -0.16% | -0.06% |
| Age $\times$ Surface Type | 0.08% | 0.06% | -0.18% | -0.27% | -0.33% |
| Advertiser Size | 0.06% | 0.04% | -0.15% | -0.14% | -0.27% |

## 4.2 EVALUATION METRICS

Our evaluation centers on the primary metric for model performance in large-scale ranking systems: Normalized Entropy (NE). NE measures the model's binary cross-entropy, normalized by the entropy of a baseline model that predicts the global average event rate. A lower NE indicates better performance. It is defined as:

$$\text{NE} = \frac{-\frac{1}{N} \sum_{i=1}^{N} \left[ y_i \log(\hat{y}_i) + (1 - y_i) \log(1 - \hat{y}_i) \right]}{-\frac{1}{N} \sum_{i=1}^{N} \left[ y_i \log(\bar{y}) + (1 - y_i) \log(1 - \bar{y}) \right]} \tag{13}$$

where $y_i$ is the true label for the $i^{\text{th}}$ sample, $\hat{y}_i$ is the predicted probability, and $\bar{y} = \frac{1}{N} \sum_{i=1}^{N} y_i$ is the empirical label mean. We report the relative improvement as $\text{NE}_{\text{diff}} = \frac{\text{NE}_{\text{C2AL}} - \text{NE}_{\text{baseline}}}{\text{NE}_{\text{baseline}}}$, evaluated on both the overall set and on the specific minority cohorts used for auxiliary task construction.

Our analysis of the results is partitioned by the model's optimization objective. We first examine C2AL's impact on six production models, which have fundamentally distinct signal characteristics covering engagement and conversions.

## 4.3 EXPERIMENTAL RESULTS

In our experiments, we cover models which optimize for different ranking funnel stages, optimization objectives, platform and surface types, as mentioned in Section 4.1.

As detailed in Table 2, applying C2AL yields statistically significant reductions in NE. Improvements in these foundational models carry substantial downstream impact across the entire ads ecosystem. In production environments of this scale, offline NE reductions of this magnitude correspond to substantial online gains and improvement in the value delivered to both advertisers and users.

Table 2: Overall performance of C2AL compared to the baseline across six production models. C2AL consistently improves predictive accuracy. Model A and B optimize for the click objective, while Models C to F optimize for the conversion objectives. Lower NE$_{\text{diff}}$ values indicate better performance.

| | Model A | Model B | Model C | Model D | Model E | Model F |
|---|---|---|---|---|---|---|
| NE$_{\text{diff}}$ | -0.07% | -0.11% | -0.16% | -0.15% | -0.08% | -0.05% |

To understand the source of these gains, we analyze the performance of the Model A at a finer granularity. The C2AL cohorts were constructed using the bottom 5% (p0-p5) and top 5% (p95+) of users, defined by a proprietary user-value metric, as the contrastive cohorts. Figure 6 reveals that C2AL not only improves performance on these targeted tail (-0.19% NE$_{\text{diff}}$) and head (-0.05% NE$_{\text{diff}}$) segments but also generalizes across intermediate user segments. Notably, the gains are most pronounced for high-value users (p75+), with NE reductions exceeding 0.04%. This disproportionate improvement is expected, as the behavior (e.g., PLR) of high-value segments often diverges most significantly from the population mean, much more so than that of low-value users—making them prime beneficiaries of a better attention mechanism that counteracts majority bias. C2AL effectively regularizes

the model to better capture patterns within important minority segments without sacrificing, and indeed while enhancing, performance on other valuable user sub-populations.

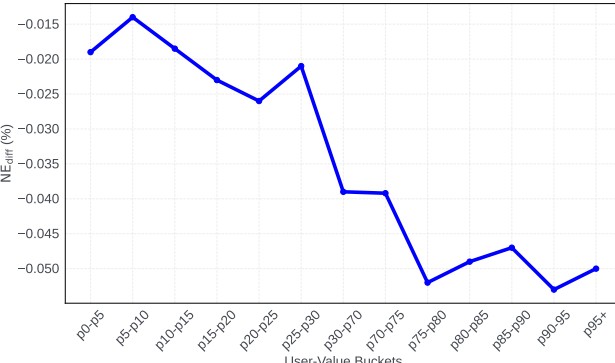

Figure 6: $NE_{diff}$ performance across user-value segments on the Model A (optimize for click objective on Instagram) with C2AL. Improvements are observed not only in the head and tail cohorts but also across intermediate segments, particularly high-value cohorts. Lower $NE_{diff}$ values indicate greater performance improvements.

We next evaluate C2AL on four conversion models, which optimize for a significantly sparser signal than clicks. The results in Table 2 again show consistent NE improvements across all four models. We observe larger relative gains for the onsite conversion models, Model C and Model D, which achieve NE reductions of 0.16% and 0.15%, respectively, while the offsite conversion models, Model E and Model F, also benefit, with NE improvements of 0.08% and 0.05%. The results suggest that C2AL is particularly effective for onsite conversion prediction, where user journeys are more completely observed.

To probe the robustness of these improvements, we analyze the Model C's performance when contrastive cohorts are defined along four distinct semantic axes (Table 1) . C2AL delivers consistent overall gains regardless of the cohort definition, with overall $NE_{diff}$ ranging from -0.14% (Age) to -0.26% (Revenue). Crucially, the overall improvement is not simply a weighted average of the gains in the head and tail segments. For instance, under the "age" axis, the overall NE improves by 0.14% despite an only 0.06% reduction in the tail. This demonstrates that C2AL enhances the model's fundamental representations, leading to broad-based generalization improvements across the entire data distribution, rather than merely overfitting to the targeted sub-populations.

## 5 CONCLUSION

In this work, we addressed the problem of latent cohort under-representation in large-scale recommendation models, where global optimization leads to representations biased toward majority populations, degrading performance on certain minority cohorts. We introduced Cohort-Contrastive Auxiliary Learning (C2AL), a practical and scalable framework that mitigates this bias. By systematically discovering axes of high predictive contrast and formulating targeted auxiliary tasks, C2AL enables the model to learn more robust and cohort-aware representations at no additional inference cost.

Our evaluation on six distinct, global-scale production ads models demonstrates that C2AL yields consistent and significant improvements in predictive accuracy. A key contribution of our work is moving beyond empirical results to provide a clear mechanistic link between the C2AL objective and its success. We showed empirically and mathematically how the cohort-contrastive tasks modulate the model's internal attention mechanisms to promote a denser, higher-diversity weight distribution, leading to richer and more diverse feature interactions. The resulting improvements are not confined to the targeted cohorts but generalize across the entire data distribution, underscoring a fundamental enhancement in representation quality. C2AL offers an interpretable approach to mitigating representation bias, and we believe this paradigm of targeted, contrastive regularization holds significant promise for a wide range of industrial-scale machine learning applications.

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

# A APPENDIX

## A.1 THE USE OF LARGE LANGUAGE MODELS (LLMS)

We leveraged large language models (LLMs) exclusively for linguistic enhancement throughout our process. Their role was strictly limited to tasks such as identifying and correcting grammatical errors, and polishing the language. At no point did we utilize LLMs for generating original content, conducting analysis, or influencing the substantive ideas presented. This careful and focused application of LLMs underscores our commitment to maintaining the integrity and authenticity of our content while benefiting from advanced linguistic tools.

