# OpenReview forum: "C2AL: Cohort-Contrastive Auxiliary Learning For Large-Scale Recommendation Systems"
_ICLR.cc/2026/Conference — Submitted to ICLR 2026_

### Official Review · Reviewer_bcT9 · 2025-10-25

**Soundness:** 3
**Presentation:** 3
**Contribution:** 2
**Rating:** 4
**Confidence:** 4

**Summary:**

This paper proposes Cohort-Contrastive Auxiliary Learning (C2AL), a framework to mitigate representation bias in large-scale recommender systems. By identifying user cohorts with high predictive divergence and introducing contrastive auxiliary tasks during training, C2AL enhances attention diversity and improves minority cohort representation.

**Strengths:**

1. It introduces a novel cohort-contrastive auxiliary learning approach that bridges multi-task learning and bias correction in recommendation systems.
2. The paper provides clear gradient-based analysis linking auxiliary tasks to improved attention diversity and representation learning and offers interpretable insights through weight distribution and training dynamics visualization, enhancing model transparency.
3. Applies easily to large-scale models with no additional inference cost, making it highly practical for industrial deployment.

**Weaknesses:**

1. A key limitation of C2AL is its reliance on manually defined semantic axes for cohort segmentation (e.g., user age or advertiser size), without a clear or principled selection procedure. This manual choice introduces subjectivity and may lead to inconsistent results across datasets or platforms. Moreover, the method only leverages the two cohorts with the largest distributional divergence (“head” and “tail”), but the rationale for using exactly two is not well justified. Real-world data often exhibit more complex subgroup structures, and restricting to a binary contrast may oversimplify the heterogeneity.

2. While the performance gains achieved by C2AL are statistically significant, the improvements are relatively small in absolute terms (around 0.1% in normalized entropy).

3. In Figure 2, it is unclear whether the left and right plots correspond to experiments on different datasets or different model settings. While the visualization idea in Figures 2 and 4 is appealing, the baseline (blue) curves are largely obscured by the red ones, making it difficult to distinguish peak density and distribution width.

4.The innovation needs further clarification. According to Equation (4), the method mainly separates head and tail data for modeling and introduces a regularization term in the latent space. This is a common approach for handling data imbalance. The authors should more explicitly articulate how C2AL differs from or improves upon existing latent-space alignment and auxiliary learning methods.

**Questions:**

1. The paper does not provide a detailed analysis of how the auxiliary loss weights (λ_head and λ_tail) affect model behavior or performance. These hyperparameters likely play a crucial role in balancing the influence of primary and auxiliary gradients, especially under partially conflicting objectives.

2. It remains unclear whether the observed improvements genuinely arise from using the two most divergent cohorts or simply from the introduction of auxiliary learning itself. If cohorts with smaller or random divergence were used, would the model still exhibit similar attention weight diversification? This ambiguity raises questions about whether the gains stem from cohort-contrastive design or merely from additional regularization.

---

### Official Review · Reviewer_LxWz · 2025-10-26

**Soundness:** 2
**Presentation:** 2
**Contribution:** 1
**Rating:** 2
**Confidence:** 3

**Summary:**

This paper addresses the problem of data heterogeneity in recommender systems leading to model bias toward the majority cohort and proposes a cohort contrastive auxiliary learning framework, C2AL. This framework first automatically discovers "head" and "tail" cohorts with high predictive distribution disparity and then constructs cohort-specific auxiliary classification tasks. The gradient signals from these auxiliary tasks serve as a functional regularizer, forcing the model's attention mechanism to learn denser and more diverse weight distributions.

**Strengths:**

1. C2AL addresses the problem of model bias toward the majority cohort.
2. Gradient analysis demonstrates how the auxiliary loss injects cohort-specific gradient signals into the attention matrix.
3. Experiments are conducted on a large-scale dataset.

**Weaknesses:**

1. Although Figure 2 demonstrates the sparsity of attention weights, this does not prove that the selection of these features is driven by the majority cohort. A more detailed division of features into majority and minority cohorts should be performed.
2. The motivation for constructing an auxiliary task on the pair of cohorts with maximal distributional disparity is unclear.
3. The paper indicates the use of multiple distributional divergence metrics (such as KL divergence and Cosine similarity) to quantify the difference, but does not report how these metrics were chosen in the experiments.
4. The performance improvement is not significant. Although the authors emphasize the significance of this improvement, no significance test is performed.
5. The lack of an experimental setup makes the reproducibility of the proposed algorithm unconvincing.
6. The lack of sensitivity analysis, for example, how the hyperparameters introduced in Equation 10 affect performance.
7. The lack of ablation experiments, for example, to analyze the performance of using only one auxiliary task.
8. The paper targets large-scale recommender systems, but there is no specific design for such systems; it appears to be just one application.

**Questions:**

Please refer to Weaknesses.

---

### Official Review · Reviewer_HrEm · 2025-10-29

**Soundness:** 2
**Presentation:** 3
**Contribution:** 3
**Rating:** 4
**Confidence:** 4

**Summary:**

This work argues that user data in  RS exhibit heterogeneous distributions. A single learning objective may cause the model to focus mainly on the behavioral representations of majority users while neglecting long-tail users, leading to poorer accuracy for the latter.
To address this, the authors propose a contrastive auxiliary task during training. Experiments on industrial-scale datasets and online A/B testing demonstrate that the proposed C2AL method is effective.

**Strengths:**

- The research problem is valuable, as it focuses on the heterogeneous distribution of users.
- C2AL is applied only during the training stage, resulting in minimal additional cost.
- The effectiveness of the method is validated through online A/B testing.
- The paper provides visualization analyses, examining the attention weight distribution to demonstrate the effectiveness of  C2AL.

**Weaknesses:**

- The proposed method relies on manually defined priors to label and divide user cohorts, rather than an automated approach, which makes the manually set hyperparameters potentially influential.
- Even among long-tail users, not all samples are truly “unique,” so simply partitioning users into groups may be an oversimplified and potentially unreasonable assumption.
- The paper does not consider the optimization conflict between the auxiliary and primary tasks, and the hyperparameter λ still requires manual tuning, which may introduce risks.
- Common advertising metrics such as AUC are not reported, even though I understand that NE is an important metric.
- The work lacks validation results on public datasets.

**Questions:**

1. In online advertising, multiple tasks are often trained jointly rather than optimized independently; in such cases, how to design labels for auxiliary tasks remains unclear.
2. Is there any analysis of user embedding of long-tail users?
3. Did you consider any multi-embedding method[1,2,3] to avoid conflict representions?
4. Is the method still suitable for transformer-based recommendation model, e.g., [4]?

[1] STEM: unleashing the power of embeddings for multi-task recommendation. AAAI 2024.
[2] Ads Recommendation in a Collapsed and Entangled World. KDD 2024.
[3] On the Embedding Collapse when Scaling up Recommendation Models. ICML 2024.
[4] RankMixer: Scaling Up Ranking Models in Industrial Recommenders.

---

### Official Review · Reviewer_Xmm1 · 2025-11-01

**Soundness:** 2
**Presentation:** 2
**Contribution:** 2
**Rating:** 4
**Confidence:** 4

**Summary:**

This paper builds on the DHEN architecture for multi-task recommendation and adds a cohort-level contrastive auxiliary learning idea. Basically, it introduces cohort-based auxiliary heads and a contrastive regularization term to make the shared representation more discriminative across different user or ad groups. The idea is quite straightforward — using group-level contrast instead of instance-level contrast — and it’s a reasonable extension of DHEN. The paper also gives some gradient analysis to explain how the auxiliary loss affects shared features. Overall, it’s a simple and practical modification with clear motivation, though not a big conceptual jump.

**Strengths:**

The paper starts from a reasonable motivation — improving the DHEN framework for multi-task recommendation — and the idea is clear.

Introducing cohort-level contrastive auxiliary learning is a natural and practically meaningful direction to enhance representation learning.

The method is simple, easy to implement, and can be smoothly integrated into existing systems.

The gradient decomposition analysis helps explain the intuition of reducing task interference.

**Weaknesses:**

The novelty is limited. The idea of cohort-level contrast or group-based regularization has already appeared in prior work (e.g., fairness or group-robust learning). Here it feels more like an extension on top of DHEN rather than a new concept.

The so-called contrastive loss is closer to an auxiliary classification objective and doesn’t really exploit the core property of contrastive learning.

Experiments are done on large industrial data but lack strong baselines (e.g., PLE, MMoE, GroupDRO, PCGrad, etc.), making it hard to tell whether the improvement is substantial.

No ablation studies are provided to show how different components (e.g., cohort definitions or loss weights) contribute to the result.

The reported improvements (0.05%–0.16%) are very small, and there’s no statistical significance analysis, which weakens the empirical claims.

The theoretical part is more interpretive than rigorous, without real theoretical innovation.

**Questions:**

The paper motivates cohort-level contrastive learning as a way to capture heterogeneity among different user groups, but the experiments mainly focus on long-tail splits or data imbalance.
Could the authors clarify why there is this mismatch between the motivation and the experimental design?
Why not explore more cohort dimensions (e.g., user interests, regions, device types) to demonstrate the method’s effectiveness under broader forms of heterogeneity?

---

### Meta-Review · Area_Chair_32To · 2025-12-29

**Summary:**

The reviewers' concerns primarily focused on the contribution and experiments presented in this work. For the contribution, reviewers are concerned that the handcrafted work may affect generalization, and the novelty is unclear. For experiments, many details are lacking, and the improvements are not significant. And more public results are suggested to validate the method. The work is not yet ready for publication.

**Reviewer Concerns:**

Since the authors have not provided a rebuttal, the contribution and experiments remain outstanding issues in this work. The manual work involved will severely affect generalization, and the novelty is limited. The experimental details should be enhanced.

**Reviewer Scores:**

Since the authors have not provided a rebuttal, I believe that all the reviewers will keep their scores.

---

### Decision · Program_Chairs · 2026-01-26

Reject